# mRNA COVID-19 Vaccines and Long-Lived Plasma Cells: A Complicated Relationship

**DOI:** 10.3390/vaccines9121503

**Published:** 2021-12-20

**Authors:** Girolamo Giannotta, Nicola Giannotta

**Affiliations:** 1Azienda Sanitaria Provinciale Vibo Valentia, 89900 Vibo Valentia, Italy; 2Medical and Surgery Sciences, Faculty of Medicine, Magna Græcia University, 88100 Catanzaro, Italy; nicola.giannotta@studenti.unicz.it

**Keywords:** SARS-CoV-2, SARS-CoV-2 mRNA vaccines, COVID-19 vaccines, immune memory after COVID-19 disease, immune memory after COVID-19 vaccination, long-lived plasma cells after mRNA COVID-19 vaccines

## Abstract

mRNA COVID-19 vaccines have hegemonized the world market, and their administration to the population promises to stop the pandemic. However, the waning of the humoral immune response, which does not seem to last so many months after the completion of the vaccination program, has led us to study the molecular immunological mechanisms of waning immunity in the case of mRNA COVID-19 vaccines. We consulted the published scientific literature and from the few articles we found, we were convinced that there is an immunological memory problem after vaccination. Although mRNA vaccines have been demonstrated to induce antigen-specific memory B cells (MBCs) in the human population, there is no evidence that these vaccines induce the production of long-lived plasma cells (LLPCs), in a SARS-CoV-2 virus naïve population. This obstacle, in our point of view, is caused by the presence, in almost all subjects, of a cellular T and B cross-reactive memory produced during past exposures to the common cold coronaviruses. Due to this interference, it is difficult for a vaccination with the Spike protein alone, without adjuvants capable of prolonging the late phase of the generation of the immunological memory, to be able to determine the production of protective LLPCs. This would explain the possibility of previously and completely vaccinated subjects to become infected, already 4–6 months after the completion of the vaccination cycle.

## 1. Introduction

The current pandemic caused by SARS-CoV-2 has been a powerful input towards scientific research, and a substantial proportion of this research has involved vaccinology. Due to the state of emergency, the regular phases of vaccine studies, and their timing, have been accelerated. As a result, some issues only emerged after billions of doses of vaccine were administered around the world.

The fundamental problem that is the focus of our attention is the evanescence of humoral immunity produced by vaccination. In fact, most immunological studies have shown that protection against infection does not last more than 4–6 months after the vaccine schedule is completed.

It was interesting to try to understand why previously vaccinated subjects would subsequently become infected, and possibly sick. We studied the molecular and cellular mechanisms of waning immunity that follows the injection of mRNA vaccines.

Excluding pandemic effects driven by viral variants, we hypothesize that the two phenomena of waning immunity and non-sterilizing immunity have a single origin that lies in the characteristics of these vaccines, which appear to be able to evoke MBCs but not able to evoke LLPCs in subjects naive to SARS-CoV-2 virus.

All of this has important implications for the question of whether the pandemic can be effectively controlled with these vaccines alone, due to the possibility of breakthrough SARS-CoV-2 infections in vaccinated subjects.

## 2. Mechanism of Action of mRNA Vaccine

In both mRNA vaccines, the mRNA molecule consists of the following elements: a 5’cap attached to the 5’ UTR, which is followed by the coding sequence for the SARS-CoV-2 spike protein; a 3’ UTR, and a long poly-A tail [1,2]. The coding sequences of mRNA vaccines are composed of the viral spike protein encoding mRNA, incorporated in a lipid nanoparticle (LNP) and stabilized by polyethylene glycol (PEG). LNPs enter target cells after they overcome the obstacles of the cell membrane, and after uptake, they are found in endosomal vesicles. From these vesicles a small portion of mRNA passes into the cell’s cytoplasm for translation [3]. 

The fact sheet for healthcare providers administering vaccines reports this mechanism of action of the Pfizer vaccine (point 14.1):“The modRNA in the Pfizer-BioNTech COVID-19 Vaccine is formulated in lipid particles, which enable delivery of the RNA into host cells to allow expression of the SARS-CoV-2 S antigen. The vaccine elicits an immune response to the S antigen, which protects against COVID-19” [4]. The nucleoside-modified mRNA in the Moderna COVID-19 Vaccine “is formulated in lipid particles, which enable delivery of the nucleoside-modified mRNA into host cells to allow expression of the SARS-CoV-2 S antigen. The vaccine elicits an immune response to the S antigen, which protects against COVID-19” [5]. It is evident from these data that the phrase “delivery of the RNA into host cells” is not the mechanism of action of the vaccine.

The innate immune mechanisms by which mRNA vaccines generate potent adaptive immunity are widely unknown. However, the mRNA vaccines are taken up by antigen-presenting cells (APCs) via endocytic pathways, after intramuscular injection. Monocytes and dendritic cells (DCs) at injection sites and draining lymph nodes (LNs) showed adequate LNP uptake and mRNA translation [6]. Cationic lipids are considered to enhance RNA uptake and subsequent exit from endosomes [7]. An experiment conducted with cultured dendritic cells (DC_2.4_) showed that these cells capture certain LPNs and exhibit a significantly higher transfection efficiency [8]. In any case, any vaccine administration results in a strong production of proinflammatory cytokines, together with a transient infiltration of APCs to the LNP/mRNA-injected site. To develop an antigen-specific immune response, an mRNA vaccine must reach the cytosol of recipient cells and express the antigen. After intramuscular vaccination, antigen expression occurs mainly in myocytes; however bone marrow-derived APCs are required to prime TCD8 cells using class I major histocompatibility complex (MHC) molecules [9]. Furthermore, the internalization of mRNA vaccines also involves non-immune cells at the injection site. After the injection of exogenous mRNA, the molecules can be internalized in non-immune cells, generating up-regulation of cytokines and chemokines at the immunization site [10]. The expression of the antigen in these non-immune cells can cause the triggering of antigen-specific antibodies and the induction of TCD8 lymphocyte responses through cross-presentation [11]. TCD8 cells, specific for pre-existing and/or infection-induced SARS-CoV-2, are the main determinants of immune protection at the individual and population level [12]. For this reason, it is important that vaccines evoke specific TCD8 responses that are lasting over time, also in light of the fact that humoral immunity is evanescent and signals that these vaccines hardly evoke the production of LLPCs, which are responsible for the defense against infection. Overall, the absence of LLPCs is documented by the absence of spike-specific antibodies 4–6 months after the second dose of vaccine, although in some studies low antibody titers persist beyond 6 months. After infection, the memory B cells (MBCs) and TCD8 lymphocytes must intervene to block the progression of the infection. It has been shown that vaccination with an mRNA vaccine results in the production of TCD8 cells [13], but these cells were capable of expressing a lower concentration of CD38 molecules than natural infection.

The activation of naive TCD8 cells is best triggered by antigen presented by DCs. The antigen that can activate them must be in the cytosol, in order to be complexed with the class-I MHC molecules. Therefore, the vaccine spike protein must be present in the cytosol of the DCs that have captured the LNPs in the extracellular area, or from the non-immune cells, at the injection site. After exiting the endosomes, mRNA can be translated in the ribosomes and then arrives at the endoplasmic reticulum (ER), undergoes post-translational modifications [14], and can be associated with Class-I MHC molecules, and then exposed to the DC surfaces for presentation to B and TCD8 lymphocytes. 

Now it remains to be understood how TCD4 helper lymphocytes can be activated. BNT162b1 induces proinflammatory and functional TCD4 and TCD8 cell responses in almost all participants, with T_H1_ polarization of the helper response [15], but BNT162b1 is a secreted and truncated version of the spike protein consisting of the RBD region, and the immunogenicity study was based on the RBD-binding IgG concentrations and SARS-CoV-2-neutralizing titer assay [16]. Kalimuddin et al. [17] found S-reactive TCD4 and TCD8 cells as early as day 7 and day 10 after vaccination with an mRNA vaccine. However, cross-reactive or infection-induced SARS-CoV-2 specific TCD8 cell responses are relevant for immune protection in mild SARS-CoV-2 infection [12]. Conceptually, T cells do not recognize the virus [18], but only the virus-infected cells (TCD8 cells), or cells that have internalized viral antigens produced after infection (TCD4). So, infection is not prevented by T cells; only LLPCs have this ability.

## 3. From Deltoid Muscle to Axillary Lymph Nodes

Positron emission tomography with computed tomography (PET/CT) scans of recently vaccinated people showed increased uptake in the deltoid muscle, corresponding to the vaccine injection site, as well as in the ipsilateral (enlarged) axillary lymph nodes. In this study, the authors were able to show that 45% of patients demonstrate avid ipsilateral axillary lymphadenopathy on FDG PET/CT in the weeks following vaccination with the novel mRNA-based COVID-19 vaccine [19]. The fact that less than half of vaccinated subjects exhibit this reactivity may be related to the clearance of LNPs, the possible presence of anti-PEG antibodies, the amount of antigen that comes out of the endosomes, and other factors of the individual immune response. Avid ipsilateral lymphadenopathy indicates that the immune response to the recently injected vaccine is concentrated in the ipsilateral axillary lymph nodes. Given this peculiarity, patients who have undergone treatment for unilateral breast cancer should be vaccinated in the contralateral deltoid to avoid unnecessary lymph node biopsies. There is also clinical confirmation of an ipsilateral lymphadenopathy following vaccination with mRNA-based vaccines [20], and two case reports of necrotizing histiocytic lymphadenitis (Kikuchi—Fujimoto Disease) after Pfizer/BioNTech COVID-19 vaccine injection [21].

In the axillary lymph nodes, DCs and T and B cells specific for the captured antigen must be encountered. DCs inside the injection site (deltoid muscle) capture the LNPs that contain viral spike protein encoding mRNA, and the spike protein that is expressed by other cells, and then transport them to draining lymph nodes. During the migration, the DCs mature and become efficient APCs. Now, the DCs express the CCR7 receptor that is specific for two chemokines, CCL19 and CCL21, which are produced in the T cell zones of lymph nodes and in lymphatic vessels. Naive T cells recirculating through lymph nodes encounter these APCs, and the T cells that are specific for the displayed peptide-MHC complexes are activated. Naive T cells also express CCR7 and migrate to lymph nodes, precisely to the same region where antigen-bearing DCs are concentrated. The same antigen activates naive B cells in the follicles, and now these cells must meet with the helper T cells activated by the same antigen. Activated T cells then downregulate receptor CCR7 and increase the expression of CXCR5 and, as a consequence, leave the T cell zone and migrate toward the follicle. The CXCR5 receptor ligand is CXCL13, which is produced by follicular dendritic cells (FDCs) and other follicular stromal cells. B cells respond to that antigen, reducing the cell surface expression of the chemokine receptor CXCR5 and increasing expression of CCR7. As a result, activated B cells migrate toward the T cell zone, drawn by a gradient of CCL19 and CCL21, which are the ligands of CCR7. The pure result of these changes is that antigen-activated T and B lymphocytes are drawn towards each other and make contact with the edge of primary follicles [22].

## 4. Protein Spike Sequence in mRNA Vaccines

Shortly after the release of the viral genome sequence, an ectodomain of the spike protein (named S-2P) was engineered to stabilize the conformation to the pre-fusion state. This S-2P ectodomain comprises the first 1208 residues of the spike protein to which two proline stabilizing mutations in the S2 domain have been associated and a trimerization motif has been added at the C-terminus [23]. BNT162b2 and mRNA-1273, which are, respectively, the Pfizer/BioNtech and Moderna anti-SARS-CoV-2 mRNA vaccines, both encode the full-length spike protein. 

## 5. Spike’s Alternate Forms

Unfortunately, the canonical version of the spike protein, the one present in vaccines, is one of the possible alternatives that presents the drawback of masking some important immunogenic epitopes conserved among coronaviruses in the RBD domain [24].

## 6. Processing of Antigens

Protein antigens can be processed two different ways: (1) the antigens synthesized within the cell are in the cytosol and generate class I-associated peptides, which are recognized by TCD8 cells; (2) antigens that are internalized from the extracellular environment into the vesicles of APCs usually generate peptides that are displayed by class II MHC molecules and recognized by TCD4 cells. MHC molecules are required to provide antigens to T lymphocytes. Class I molecules are expressed virtually on all nucleated cells, whereas class II molecules are expressed on DCs, B lymphocytes, macrophages, and a few other cell types only [25]. Class I MCH and its peptide create a complex ligand that acts as the target of the T-cell receptor (TCR) on the T-cell surface [26]. Class II peptide loading occurs in coordination with phagocytosis and lysosomes. If part of the spike protein produced after vaccination becomes extracellular antigen, it is captured and internalized into endocytic vesicles of APCs. These vesicles merge with the lysosomes forming endosomal/lysosomal vesicles where the vaccine antigen processing takes place. After the proteolytic degradation of internalized proteins in the endocytic vesicles, peptides are bound to class II MHC molecules in vesicles. Class II MHC molecules are synthesized in the ER and transported to endosomes, where the peptide-class II MHC complexes are assembled and are now ready to be expressed on cell surfaces of the APCs; here, they are displayed for recognition by TCD4 cells.

Conversely, antigen expression in non-immune cells can induce the responses of TCD8 lymphocytes by cross-presentation [11]. Alternatively, the presence of the spike protein expressed in the cytosol of DCs makes it possible to use the intracellular pathway of presentation of antigens that leads to their association with class I MHC molecules, then presentation to TCD8 lymphocytes. It is likely that the DCs express the spike protein by translating the LNP mRNA, and at this point, they should have the antigen to present in the cytosol. 

For the initiation of adaptive immune responses, DCs present antigenic peptides, in association with class II MHC molecules, to naïve TCD4 lymphocytes for interaction with the T-cell receptor (TCR) [27,28]. Antigen uptake is closely followed by the upregulation of class I MHC, class II MHC, and co-stimulatory molecules on the surface of DCs [29]. Peptides can be generated either by lysosomal proteases in the endocytic pathway, or by proteasomes, when endocytosed proteins are transferred through the endosomal membrane into the cytosol. Thus, generated peptides may associate intracellularly with either class I MHC (MHC-I) or class II MHC (MHC-II) molecules, and, in that context, they can be transferred to and displayed at the plasma membrane [27].

## 7. CD Molecules

Cluster of differentiation (CD) antigens consist of a series of molecules that are expressed on the lymphocyte membranes and that allow us to understand which cell is being talked about in vaccine efficacy studies, which in addition to antigen-specific antibodies, often analyze the cell populations activated by vaccination (the immune phenotypes). “All B lymphocytes are CD19^+^CD3^−^; so, they express the CD19 molecule virtually on all human cells of the B-lymphocyte lineage [30], except BM LLPCs [31]”. In humans, the BM CD19^−^CD38^high^CD138^+^ subset is the main component of the LLPC pool [32]. 

### 7.1. CD27

Originally [33], the surface marker CD27 was used to differentiate memory (CD27^+^) from naive (CD27^−^) B cells. Subsequently, a subset of MBCs (CD27^−^IgD^−^) was also found to be negative for both CD27 and IgD markers [34]. Naive B cells (CD19^+^CD3^−^CD27^−^IgD^+^) can be IgD^+,^ while the subset of MBCs is negative for this marker (CD19^+^CD3^−^CD27^−^IgD^−^). In addition, there are two other subsets of CD27^+^ MBC with IgD^+^ or IgD^−^.

### 7.2. CD38

When the antigen arrives in the lymph node, towards which a primary response has been developed, the B cells react quickly and energetically, producing a secondary response that affects both the MBCs and the naive cells that have a certain reactivity against the same antigen. In lymphoid follicles, some B lymphocytes express the CD38 marker, and the part located within the germinal centers (GCs) is composed of two groups of cells that are in small part progenitors, and in large part, activated B cells [35].

In other words, in a secondary antigenic challenge, MBCs are ready to respond quickly and with great efficiency, while naive cells take longer to provide adequate humoral responses. It is possible to distinguish recently activated naive B cells (CD27*^low^*CD38*^low^*CD20*^high^*) from memory-derived activated B cells (CD27*^high^*CD38*^high^*CD20*^low^*) ex vivo, when assessing acute responses to infection or vaccination [36].

CD38 is positive in plasma cells and is a marker of cell activation. CD27 is positive in MBCs, while it is low in recently activated naive B cells. CD20 is low in MBCs, while it is high in recently activated naive B cells. Blimp-1 in the germinal center (GC) is associated with an intermediate phenotype between GC B cells and plasma cells. CD71 is expressed in all proliferating cells. A fraction of plasmablasts (CD3^−^CD19^+^CD20^−^IgD^−^ CD27*^high^*CD38*^high^*), migrates from GC to the BM, where they become LLPCs within the survival niches [37]. Naive B cells express only IgM and IgD; CD27 markers are expressed by MBCs. 

## 8. Blimp-1

B lymphocyte-induced maturation protein-1 (Blimp-1) serves as a master regulator of both the development and function of antibody-producing B cells. Blimp-1 was shown to promote B-cell terminal differentiation. Forced expression of Blimp-1 drives mature B cells to differentiate into plasma cells [38]. Blimp-1 was not detected in MBCs. The expression of Blimp-1 in the GC is, thus, associated with an interesting subset of cells with a phenotype that is an intermediate between the B cells of the GC and the plasma cells. Ultimately, Blimp-1 is permissive for plasma cell development and commits B cells to plasma cell fate [22,39].

## 9. B Cell Activation

The activation of the B cell starts with recognition of the specific antigen that binds to its BCR receptor. B cells are able to bind directly soluble or bound antigen from APCs. Small antigens can reach the lymph nodes after being transported by afferent lymphatics. When the lymph reaches the subcapsular site, the small molecules can channel themselves into the ducts and thus reach the follicle. If the soluble molecule is larger, it can be captured by macrophages, in subcapsular sinus, and by DCs, in medulla. With immunization, this is not the activation pathway because the antigen contained in the vaccine is transported by the DCs from the injection site to the lymph node.

### 9.1. Extrafollicular B Cell Activation

B cell activation in extrafollicular focus provides an early antibody response that produces low-affinity antibodies. The extrafollicular response also helps to generate follicular helper T cells (T_FH_ cells) that migrate into the follicle and that are required for GC formation. Furthermore, some antigen-activated B cells return to the follicle from the extrafollicular focus and participate in GC formation.

### 9.2. Induction of Follicular Helper T Cells

T_FH_ cells are a subset of the TCD4 helper cells that are involved in the regulation and development of antigen-specific B cell immunity. By activation, naïve TCD4 cells can differentiate into distinct subsets of effector Th cells with various functions [40,41,42,43,44,45]. Within 4 to 7 days after antigen exposure, activated antigen-specific B cells induce some previously activated T cells to differentiate into T_FH_ cells. T_FH_ cell differentiation from naive TCD4 cells requires an initial activation by APCs, and a subsequent activation by B cells. T_FH_ cells express high levels of the chemokine receptor CXCR5, and they are drawn into lymphoid follicles by CXCL13 (ligand for CXCR5).

### 9.3. Germinal Center Formation

T cell-dependent activation of follicular B cells can induce the formation of a germinal center (GC), which is the microenvironment where affinity maturation of the humoral immune response takes place [46]. Figure 1 illustrates the fate of a naive B cell that encounters the antigen in a peripheral lymph node.

A naïve B cell present in the lymph nodes and spleen, when activated for the first time by the complementary T-dependent antigen, undergoes a series of modifications that occur essentially in the germinal center (GC) of the peripheral lymphoid organs (Figure 1). In the GC, T-helper lymphocytes, follicular dendritic cells (FDCs), and antigen-activated B lymphocytes are encountered, and together they produce a primary immune response. The primary humoral response produces plasma cells, memory B cells (MBCs) and long-lived plasma cells (LLPCs). LLPCs produce antigen-specific antibodies for a long time if they gain a bone marrow (BM) niche that hosts them, and have the task of preventing subsequent infection in the case of a new antigenic exposure. Conversely, in the secondary immune response, the MBCs that do not produce antibodies are activated but must first be transformed into plasma cells in order to do so, within a short period of time after contact with the specific antigen. This is why a vaccinated people who cannot develop LLPCs can become infected despite having a humoral immunological memory from MBCs.

The B cells, previously selected in the extrafollicular site, can enter a compartment of the follicle that contains a network of reticular cells (CRC); this is the dark zone (DZ). Here, the B cells proliferate and give rise to the phenomenon of somatic hypermutation (SHM). There are no follicular dendritic cells (FDCs) or T cells in this compartment. B cells in this compartment are called centroblasts, which will then move into the light zone (LZ) and be called centrocytes. 

In the LZ are present T_FH_ lymphocytes in the context of a network of FDCs. FDCs have the ability to retain antigen for a long time and they have receptors for the Fc fragment of immunoglobulins and for the complement. Here, the antigen is not free but is retained by these cells in the form of an immune complex. In essence, FDCs function as APCs to B cells selected by antigen specificity, and antigen-specific B cells will also become APCs as they internalize, process, and present the antigen, in association with class II MHC molecules, to T_FH_ lymphocytes [47]. Only after all of these steps can an antigen-specific B cell become a plasma cell that produces specific antibodies or MBCs.

### 9.4. Light Zone (LZ)

In this area of the follicle, the plasmablasts can complete the final phase of their evolution, and the activated B cells complete their development in the GC by refining the antigen-specific receptor capacity and will also function as APCs, as they are able to bind and internalize more antigen and consequently present more antigen-derived peptides on class II MHCs to TCD4 lymphocytes with specific TCR (TCD4_FH_) for the same antigen. Furthermore, FDCs have antigen-antibody complexes that they expose to their cell surfaces to refine the antigen–antibody binding of high-affinity B cells. 

In other words, the selection of activated B cells favors those B cells that have a BCR with a relatively high affinity for the antigen presented by the FDCs. Furthermore, these cells will receive survival signals and can either return to the DZ, pass through another cycle of division and SHM, or exit the GC as plasma cells or MBCs [46].

## 10. Immune Memory

Vaccine effectiveness depends on immunological memory [48]. Short-lived plasma cells are generated during responses in extrafollicular B cell foci, while LLPCs are generated in T-dependent GC responses to protein antigens. MBCs are formed early in the response, whereas LLPCs are a later product [49]. Around 10% of MBCs recognize variant antigen better than wild type protein, thus allowing for a breadth of protection compared to LLPCs [50]. The plasmablasts generated in the GC enter the circulation and home to the BM, where they differentiate into LLPCs. About 3 weeks after immunization, the BM becomes a major site for antibody production. Usually, memory helper T cells (TCD4 memory cells) emerge in parallel with MBCs. 

### Long-Lived Plasma Cells

Humoral immunity that lasts over time is dependent on the LLPCs that produce IgG antibodies continuously and independently of a subsequent specific antigenic stimulus, which instead is essential for activating MBCs. Terminal differentiation of B cells generates short-lived plasma cells and LLPCs capable of producing antibodies. The transition from a short-lived plasma cell to an LLPC requires homing to the BM niche [51]. LLPCs generated in the GC of peripheral lymphoid organs (spleen and lymph nodes) during the primary immune response to a foreign antigen must be housed in survival niches found in the BM to survive. The niche is a microenvironment that provides nutritional support and survival factors such as the anti-apoptotic factor Mcl-1, which, together with the activation of the CD28 receptor of the LLPCs, helps to keep them alive. The participation of other factors to ensure this survival appears likely. In the microenvironment of the medullary niche, the following are operative: stromal cells, DCs, regulatory T cells, and other cells that produce the necessary signals and ligands such as TCD80/TCD86 for CD28. The production of soluble and stromal factors contributes to keeping LLPCs alive [52].

In the BM aspirates of 18 individuals who had recovered from COVID-19, LLPCs capable of binding the spike protein were detectable [53]. Therefore, while we are certain that COVID-19 is able to induce the production of LLPCs, we suspect that vaccinations cannot do it for several reasons that we will now present.

## 11. Cross-Reactive T Cell Immunity

The SARS-CoV-2 virus shares a broad TCD4 and TCD8 cross-reactivity with human endemic coronaviruses [54,55,56] and can evoke a secondary response by T cells to cross-reactive epitopes. The secondary response by cross-reactive T cells could immediately eradicate SARS-CoV-2 infection in a number of individuals [57,58].

## 12. TCD4 Cell-Mediated Memory

There are memory TCD4 cells (T_FH_, T_H1_, and TCD4-CTL) that can intervene in the case of a new viral infection [59]. Even not very high degrees of antigenic homology can evoke cross-reactive TCD4 cells [60]. Therefore, cross-reactive TCD4 cells (evoked by previous exposure to common coronaviruses) can be activated to limit all the determining stages of SARS-CoV-2 infection [60]. COVID-19 mRNA BNT162b2 vaccine administration also activates pre-existing spike-cross-reactive T cells [61], demonstrating that both situations (infection or vaccination) are conditioned by the presence of these spike-cross-reactive T cells. Overall, the presence of spike-cross-reactive T cells in individuals can help them to control the infection and limit its severity, although cross-reactive TCD4 cells cannot prevent infection, as this is the exclusive task of LLPCs. However, these cross-reactive T cells can also help B cells, which are committed against the SARS-CoV-2 virus, in developing an effective GC center, as they can intervene after vaccination.

In the S2 subunit of the spike, downstream of the S2’ cleavage site, there is a highly conserved cross-reactive and immunodominant peptide (S816–830) against which the reactive T lymphocytes of most infected individuals, and of almost all vaccinated individuals, are directed [62].

## 13. Cross-Reactive Immunity in Children

Children are able to develop anti-SARS-CoV-2 immunity within 7 days after the onset of symptoms. In children with COVID-19 disease, the humoral immune response is fast and efficient. Children’s B cells have converging BCRs capable of recognizing previously encountered pathogens. It could be likely that previous common coronavirus infections result in the production of MBCs, which can now be cross-reactive against SARS-CoV-2, and for this reason they can be an integral part of the anti-SARS-CoV-2 humoral defenses. A new paper confirmed that the humoral responses produced by children, after infection, correlate with the presence of SARS-CoV-2 specific TCD4 cells. Notably, individuals with elevated levels of SARS-CoV-2 specific TCD4 cells produced a potent neutralizing response that was also associated with reduced viral load [63]. It is plausible that cross-reactive MBCs and cross-reactive TCD4 cells effectively collaborate to produce a strong anti-SARS-CoV-2 humoral response. Furthermore, in adult subjects, the persistence of BCRs that are specific for the spike protein and not excessively mutated [64] may allow early virus recognition and improve the anti-SARS-CoV-2 defense mechanisms.

## 14. Re-Exposure to Spike Protein

Upon re-exposure to an antigen, the memory recall response will be faster, stronger, and more specific than a naïve response. Protective memory depends first on circulating antibodies secreted by LLPCs. If there are no LLPCs, MBCs are recalled. MBCs and/or cross-reactive MBCs contribute to the defense against the virus but cannot prevent infection. However, there is strong evidence of pre-existing cross-reactive MBCs that were activated on SARS-CoV-2 infection [65]. Even TCD4 and TCD8 memory cells can only control the infection, but they cannot prevent it because they can only recognize infected cells. Furthermore, even local immunity possible from previous infection does not last long. In other words, the possibility of avoiding SARS-CoV-2 infection depends only on LLPCs. MBCs are stationed at strategic sites where they can maximize their chance of encountering antigens. The spleen, including the marginal zone, is a major reservoir for MBCs in both mice and humans [66], as is the subcapsular sinus of lymph nodes [67].

## 15. Waning of Spike Antibody Levels

In a SARS-CoV-2 virus naïve population, there are no univocal data on the effective duration of humoral immunity, which is, however, evanescent, and which in most studies vanishes after 4–6 months [68,69,70,71,72,73], even though in one study, vaccine-evoked antibodies were detectable after 6 months from vaccination at levels of up to 7% compared to their maximum peak [74]. The waning of S antibody levels, in infection-naive individuals, was already evident 3–10 weeks after a second dose of ChAdOx1 or BNT162b2 [74]. S-RBD IgG responses after vaccination with COVID-19 mRNA vaccine show a significant initial increase in antibody levels after the second dose. However, 6 months after vaccination, these levels were reduced on average to 7% of their peak level, and this decline is somewhat expected as all vaccine-induced short-lived plasmablasts do not necessarily differentiate into long-lived plasma cells [75]. Most of the published studies and the epidemiological data that have convinced health authorities to provide a third dose, especially in Israel, have compelled us to realize that waning immunity occurs within 4–6 months of the second vaccine dose. Indeed, a nosocomial outbreak caused by the SARS-CoV-2 Delta variant occurred in a highly vaccinated population in Israel. Of the 42 infected subjects, 39 had been fully vaccinated, and of these, some became seriously ill and some died [76].

## 16. Discussion

Some vaccines demonstrate only short-lived Ab responses and protection. To be effective, a vaccine must elicit the production of efficient LLPCs. Protective memory depends first on circulating antibodies secreted by LLPCs. LLPCs emerge in the final stage of maturation of the activated B cell leaving the LZ of the GC follicle. The design of an effective vaccine must be centered on prolonging late GC responses with resultant production of more and better LLPCs [52]. LLPCs capable of binding the spike protein were detectable in BM aspirates of COVID-19 patients [53]. The first dose of vaccine in a SARS-CoV-2 virus naïve population generates primary MBCs that have not been shown to generate large receptor affinity [77]. Upon the second dose of vaccine, these primary MBCs differentiate into plasmablasts and can generate secondary MBCs that then undergo subsequent BCR maturation. Although they can help generate a GC, they still have to compete with cross-reactive MBCs that often have better receptor affinity.

We did a search on PubMed with the keywords “*long-lived plasma cells after COVID-19 vaccines*” and retrieved three papers that we analyzed in-depth. Two of these papers refer to the production of B memory after infection with SARS-CoV-2 [53,78], and one to GC that develops after a SARS-CoV-2 mRNA vaccine [79]. This latest study demonstrated that a GC is generated after receiving a SARS-CoV-2 mRNA vaccine, but also found that cross-reactive MBCs are recruited as well as newly engaged clones that target unique epitopes within SARS-CoV-2 S protein. The B cell phenotype identified in this study was CD19^+^CD3^−^IgD^*low*^CD20^*low*^CD38^+^CD71^+^BLIMP1^+^, which is programmed to become a plasma cell, but not an LLPC.

## 17. Conclusions

Because vaccinated subjects also become infected and sick, it is likely that the initial vaccination schedule is not able to elicit LLPCs and that vaccine boosters will only serve to reactivate MBCs and memory T cells evoked from the previous vaccination. Alternatively, cross-reactive MBCs and cross-reactive T cells, eventually present in vaccinated subjects, could be activated.

In other words, vaccination has a high chance of activating cross-reactive memory cells but little chance of eliciting LLPCs. The absence of LLPCs allows SARS-CoV-2 infection when the antibodies produced by plasma cells, derived from the activation of MBCs, vanish. Waning immunity seems to be already a concrete fact 4–6 months after completion of the vaccination cycle. Finally, with these vaccines, the elicited immune response is probably a secondary immune response, not a primary immune response with the natural production of LLPCs. Our hypothesis receives further support from the study by Purtha et al. [50], which discovered that about 10% of MBCs better recognize the new antigen variant than the original antigen that evoked its production.

These cross-reactive MBCs, elicited by previous infections with the common cold coronaviruses, have the ability to recognize variants of the spike protein and for this reason may contribute to developing a better immune response to SARS-CoV-2 infection and participate in the immune response to the vaccine, which appears to have more of the characteristics of a secondary than a primary response. Moreover, the only study to show that a GC is generated after vaccination with an mRNA vaccine [79] also showed that cross-reactive MBCs are recruited as well as newly engaged clones. The B cell phenotype identified in this study (CD19^+^CD3^–^IgD^*low*^CD20^*low*^CD38^+^CD71^+^BLIMP1^+^) is programmed to become a plasma cell, not LLPC. 

A further factor that should be considered is the presence, in the general population, of a high percentage of subjects (72%) who have anti-polyethylene glycol (PEG) antibodies [80]. Furthermore, Moderna employees showed, in mice, that there was a dramatic and accelerated blood clearance (ABC) of mRNA-formulated LNPs that directly activate B-1 lymphocytes, resulting in the production of anti-phosphorylcholine IgM Abs in response to initial injection. Upon repeated injections, B-2 lymphocytes also become activated and generate a classic anti–PEG adaptive humoral response. The ABC response to phosphorylcholine/LNP-encapsulated mRNA is, therefore, a combination of early B-1 lymphocyte and later B-2 lymphocyte responses [81] that could render the mRNA vaccine less effective at any time after vaccination. 

In other words, the response to vaccination is conditioned by the presence of cross-reactive MBCs and cross-reactive T cells that, however, have the drawback of the original antigenic sin. Furthermore, the eventual production of a GC sees cross-reactive T cells collaborate with naive B cells whose contact with the antigen could be hindered by the presence of MBCs that could also interfere in the production of new LLPCs. In fact, the waning of humoral immunity, which occurs within 6 months of the second vaccine dose, is attributable to the absence of LLPCs.

Ultimately, we found no human studies demonstrating that COVID-19 mRNA vaccines elicit LLPCs, whereas we found many studies demonstrating the widespread presence of cross-reactive immunological memory to SARS-CoV-2. There is only one study using C57BL/6J or BALB/cJ mice showing that LLPCs are produced after injection of a SARS-CoV-2 mRNA vaccine [82]. Unfortunately, these mice are not permissive to SARS-CoV2 infection [83]. In general, the SARS-CoV-2 virus needs to be adapted in order to infect laboratory mice [84]. Hence, the production of LLPCs in this study, which used these mice, is explained by the fact that a primary response with production of LLPCs is generated in mice infected for the first time with SARS-CoV2, whereas in human studies, cross-reactive immunity induces secondary immune responses with the activation of cross-reactive MBCs. 

Probably, these vaccines evoke MBCs but do not evoke LLPCs, and for this reason, vaccination is not able to prevent a possible SARS-CoV-2 infection. Due to the limited life expectancy of MBCs, periodic antigenic stimulation is required to regenerate this type of memory. Moreover, we think that further studies are needed to clarify the kinetics of the antibody response, which does not appear to be as expected, as is the case with other vaccines. In the meantime, we suggest increasing the antigen dose (increasing the mRNA content in the LNP) in order to overcome the limits imposed by the presence of anti-LNP antibodies and cross-reactive MBCs with a better receptor affinity. Finally, we believe that it is necessary to administer a dose of vaccine every 4 months to high-risk individuals or to frail subjects, possibly increasing the amount of antigen in the booster doses. In the meantime, studies to evaluate the possibility of adding adjuvants to these vaccines would be desirable. Some interesting theoretical data derive from studies that have analyzed the ability of peroxisome proliferator-activated receptor (PPAR) agonists to improve immunological memory [48].

## Figures and Tables

**Figure 1 vaccines-09-01503-f001:**
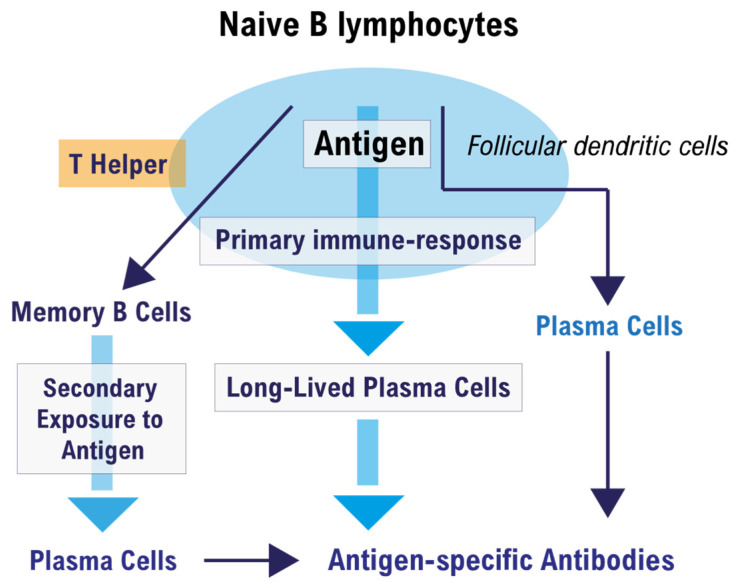
Fate of a naïve B cell after encountering the antigen in the germinal center.

## Data Availability

All the data can be requested from the corresponding author.

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
