# Peer review of "mRNA COVID-19 Vaccines and Long-Lived Plasma Cells: A Complicated Relationship"

_vaccines, 2021, doi:10.3390/vaccines9121503_

Round 1
Reviewer 1 Report
This is an interesting study concerning the complicated relationship between mRNA COVID-19 vaccines and LLPCs (long lived plasma cells).. The authors use long paragraphs concerning general and not specific knowledge of immunity to vaccines, which are not necessary in this review. Specific for COVID-19 vaccines knowledge and references should be only included. The organization of the paper is not correct. A sequence of events should be described starting from the injection to the evolution of immunity, produced by the mRNA vaccines. As it is now there is a lot of back and forth and repetitions. There are also controversies (for example it is mentioned that there are no antibodies after 4-6 months, which is not true and then it is mentioned that the antibodies are the 7% of the original. Not all the references agree on this percentage.).
Finally the conclusion is based on indirect evidence and there is no comparison to other vaccines.
Author Response
Good morning,
Thanks so much for your Comments and Suggestions.
Here are our responses.
Point 1.
This is an interesting study concerning the complicated relationship between mRNA COVID-19 vaccines and LLPCs (long lived plasma cells). The authors use long paragraphs concerning general and not specific knowledge of immunity to vaccines, which are not necessary in this review. Specific for COVID-19 vaccines knowledge and references should be only included.
Response 1 to the referee comments.
We presented the general knowledge on immunity because we felt that it was important for the reader to understand how the human body responds, in general, to the administration of vaccines. Since mRNA vaccines are not adjuvanted with aluminum salts, like many other childhood vaccines, all that knowledge cannot be applied in this case. Furthermore, we have not found data on the molecular mechanism of action of these vaccines and we had to base our work on molecular biology dates deriving from the general knowledge of the immune response to a foreign antigen.
Point 2.
The organization of the paper is not correct. A sequence of events should be described starting from the injection to the evolution of immunity, produced by the mRNA vaccines. As it is now there is a lot of back and forth and repetitions.
Response 2 to the referee comments.
We organized the study by describing the sequence of events that start from the vaccine injection site, in the deltoid muscle, and reach the axillary lymph nodes, where the immune response to vaccine administration is concentrated (See point 3 of our paper “From deltoid muscle to axillary lymph nodes”).
Point 3.
There are also controversies (for example it is mentioned that there are no antibodies after 4-6 months, which is not true and then it is mentioned that the antibodies are the 7% of the original. Not all the references agree on this percentage.).
Response 3 to the referee comments.
There was no intention of making controversy and we will correct our sentences by stating that:
- Waning of Spike-antibodies levels
There are no univocal data on the effective duration of humoral immunity which is however evanescent and which in most studies vanishes after 4-6 months, even if in one study, vaccine-evoked antibodies were detectable after 6 months from vaccination to the extent of 7%, compared to the maximum peak.
Point 4.
Finally the conclusion is based on indirect evidence and there is no comparison to other vaccines.
Response 4 to the referee comments.
Since neither the initial trials nor subsequent publications have provided molecular evidence on the mechanism of action of these vaccines, we found ourselves elaborating a putative mechanism using indirect evidence. The basic fact is that studies based on vaccine efficacy have not demostrated that LLPCs are produced after vaccine injection. Thus, there is no direct evidence that vaccination induces the production of LLPCs. Furthermore, we have not made a comparison with viral vector vaccines as the scenario is totally different. In fact, it has been shown that components of these vaccines can integrate into DNA (Doerfler W. Adenoviral Vector DNA- and SARS-CoV-2 mRNA-Based Covid-19 Vaccines: Possible Integration into the Human Genome - Are Adenoviral Genes Expressed in Vector-based Vaccines?. Virus Res. 2021;302:198466. doi:10.1016/j.virusres.2021.198466. https://www.ncbi.nlm.nih.gov/pmc/articles/PMC8168329/) and that there is the possibility of producing truncated forms of Spike due to alternative splicing (https://assets.researchsquare.com/files/rs-558954/v1/699ef798-2b1b-4375-b01d-e94a124fcb66.pdf?c=1631883403). Finally, the possibility of integration into DNA could allow some cells to express the Spike protein over time and this would represent a new antigenic stimulus that would go beyond that represented by the injection of the vaccine.
Thank you very much.
Best regards,
Dr Giannotta Girolamo.
Reviewer 2 Report
In this manuscript, the authors review the few available data regarding the antibody response duration after mRNA vaccination against the SRAS-Cov2 Spike protein. They focussed their review on the molecular mechanism of the B-cell response leading to the lack of a long-lasting immunity against SRAS-Cov2.
The authors sum up the mechanism of cation of mRNA vaccines —which basically look alike what is described in textbook— and provide putative explanation why the current mRNA vaccines cannot lead to a long lasting B-cell response.
I would suggest the authors to provide a more detailed Figure of the antibody response (presenting both cellular and molecular mechanisms) and provide some perspectives what they think that can be done to increase the immunogenicity of the mRNA vaccines.
The introduction Section should be corrected by a native English Speaker.
Author Response
Good morning,
Thanks so much for your Comments and Suggestions.
In the attached file are the responses to the reviewer.
Thank you very much.
Best regards,
Dr Giannotta Girolamo.

Reviewer 3 Report
I would applaud the authors for meticulously compiling the existing data and presenting it in a simple manner to be understood by the reader. However, I have some concerns in the manuscript:
1) Line 16- 17. The author’s hypothesis that mRNA vaccines are unable to elicit long- lasting immunological memory is not entirely correct. Please modify it accordingly. mRNA vaccines have been demonstrated to induce antigen specific memory B cell in human population (a) SARS-CoV-2 mRNA vaccines induce persistent human germinal centre responses; Nature, June 2021; (b) Evidence of SARS-CoV-2-Specific Memory B Cells Six Months After Vaccination With the BNT162b2 mRNA Vaccine; Frontiers in immunology, September 2021. However, it is true that mRNA vaccine fails to induce LLPC or there has not been enough reports.
2) Line 104:” mRNA can be translated in the ribosome” and not “mRNA can be translated into ribosome”
3) Please be consistent with CD4 and CD8 T cells. Either use CD4/CD8 T cells or use TCD4/ TCD8.
4) Section 7,8,9 contains a general information about cluster of molecules, markers for activation, germinal center reaction etc. Can these sections be compiled into one? Since these are not spoken in the context of mRNA vaccine. These sections are elongating the article, and doesn’t justify the title as they are not spoken in context of mRNA vaccine.
Author Response

(The authors gave the same response as above.)

Round 2
Reviewer 1 Report
The manuscript has been organized better and the English is better now although there are a few typos and minor mistakes. The authors insist that the general description of immune response is necessary so that the review readers understand the response to the vaccines. I wouldn't include the small sections dedicated to a few CDs (why only them and not others?)
A definite mistake is that plasma cells do not express CD19. Normal plasma cells do express CD19.
The authors mention that after 4-6 months people do not have antibodies. This is not true. The antibodies are perhaps 10% of the quantity after the second dose, but still after 9 months there are some antibodies. They should check the bibliography more thoroughly..
It is a much better organized work and it offers a lot of useful information.
Author Response
Dear Reviewer.
Many thanks for the new suggestions.
In the attached file you can find our responses and corrections made to the manuscript.
Thank you very much.
Best regards,
Dr Girolamo Giannotta.
November 21, 2021.
